# MZM Optimization of PAM-4 Transmission in Data Center Interconnect

**Eduard Sonkin [1],\*, Dan Sadot [1] and Gilad Katz [2]**

[1]   Department of Electrical and Computer Engineering, Ben-Gurion University of the Negev,
     Ness Ziona 741402, Israel; sadot@ee.bgu.ac.il
[2]   The Holon Institute of Technology, Holon 5810201, Israel; giladka@hit.ac.il
\*   Correspondence: eduard.sonkin@post.bgu.ac.il

**Abstract:** An analog optimization of 4-level pulse amplitude modulation (PAM-4) signal is proposed, together with maximum likelihood sequence estimation digital signal processing. The proposed optimizations are verified by experimental demonstration at 53 Gbaud, indicating an improvement of 4–5 dB in the optical signal to noise sensitivity.

**Keywords:** fiber optic communication; modulation format; digital signal processing (DSP); PAM–4

## 1. Introduction

The dramatic growth in capacity requirement in optical communication systems increases the motivation to encode the digital information in multilevel optical signals [1,2]. Multilevel modulation introduces lower symbol rate and narrower spectra which relax the bandwidth requirements of the optoelectronic components. On the other hand, multilevel signaling reduces the distance between the levels, resulting in lower immunity to electrical noise, optical noise, non-linear distortions, and other impairments.

One of the leading multilevel modulation techniques in the recent years for 100 Gbit/s (and beyond) and for both intra- and inter-data center optical links is 4-levelpulse amplitude modulation (PAM-4) [2–4] mainly due to its higher signal to noise ratio (SNR). Optical transmitters based on Mach–Zender Modulator (MZM) using indium phosphate and silicon photonics are becoming increasingly available for data center connections [5], especially for data centers inter-connection (DCI) which aim to connect data centers over up to 80 km [2,4,5]. Therefore, it is of great interest to introduce performance improvement techniques for PAM-4 transmission over link distance of 80 km using MZM and optical amplifiers, which is required for such fiber length. In this paper, we show two mechanisms of performance improvement to PAM-4 transmission. One mechanism of improvement is associated with the optimizing of the transmitted power levels by performing "optical uneven PAM" transmission, using the inherent non-linearity transfer function of MZM together with a non-linear equalizer, maximum likelihood sequence estimator (MLSE). The second mechanism of improvement is related to the optical modulation amplitude (OMA) improvement, utilizing the transfer function of the MZM together with a booster optical amplifier.

The remainder of this paper is divided into four sections. Section 2 presents fundamental equations associated with the power level optimization technique. Section 3 shows how MZM's transfer function is used to generate the power level and OMA optimizations. Section 4 presents lab measurements of 53 Gbaud PAM-4 (106 Gbps) optical transmission setup, while Section 5 presents the conclusions of this research. In addition, in the Appendix A we present analytical derivation of PAM-4 signal with optimum transmission levels.

## 2. Mathematical Presentation

Optical communication systems are subject to many kinds of impairments, which can be categorized mainly as: signal-independent (additive) noise, signal-dependent (multiplicative) noise, bandwidth limited induced intersymbol interface (ISI), non-linear distortions, and limitation on the signal strength such as a finite extinction ratio (ER). If signal-dependent noise dominates, higher power level is received with more noise and vice versa lower power level is received with lower noise [6,7]. Typical examples of signal-dependent noise include relative intensity noise (RIN) and amplified spontaneous emission (ASE) in optically amplified systems, and shot noise (quantum noise). The total variance associated with these noises can be written as:

$$\sigma_l^2 = \sigma_{th}^2 + \sigma_{sp-sp}^2 + \sigma_{s_l}^2 + \sigma_{sig-sp_l}^2 + \sigma_{RIN_l}^2, \; 0 \leq l \leq 3 \tag{1}$$

where $l$ is an integer that represents one of the four levels of the PAM-4 signal, $\sigma_{th}^2$, $\sigma_{sp-sp}^2$ are the thermal noise variance, and spontaneous-spontaneous beat noise variance, respectively. $\sigma_{s\,l}^2$, $\sigma_{sig-spl}^2$ and $\sigma_{RINl}^2$, are the shot noise variance, signal-spontaneous variance and RIN variance, respectively, which are multiplicative noise terms and are the dominate contributors in Equation (1), while the contribution of thermal and shot noises in many cases is neglected [8]. Therefore, higher power signals suffer from higher noise. Assuming that the noise distribution is Gaussian, the conditional probability density functions (CPDF) associated with each level is expressed by [9]:

$$p(r/s_l) = \frac{1}{\sigma_l \sqrt{2\pi}} e^{-\frac{(r-m_l)^2}{2\sigma_l^2}} \tag{2}$$

where $m_l$ is the received signal mean assuming symbol $s_l$ (level $l$) was transmitted. Figure 1 presents an example of the CPDF of a PAM-4 signal with the parameters of the lab experiment discussed in Section 4, at a bit error rate (BER) of 5e-3. In Figure 1, the mean power levels (associated with the PAM–4 symbols) are equally spaced, while the signal dependent (multiplicative) noise dominates. It is clearly observed that the CPDF of higher-level power contains higher variance as compared to lower level power. Consequently, the symbols associated with the higher levels powers are subject to more errors than the symbols associated with the lower level powers, which results in suboptimum bit error rate (BER) performance. Optimization of the BER performance in such case of signal-dependent noise may be achieved by power level spacing optimization. The optimization is done by introducing larger spacing between the higher power levels and vice versa lower spacing between the lower power levels, so that the symbols error rate (SER) at the higher levels will be similar to the SER at low levels.

In this paper we combine the power level spacing optimization in an "optical uneven PAM-4" manner, followed by electrical digital signal processing (DSP) using MLSE [3]. The DSP compensates for the non-linear distortions and ISI introduced by the band limited MZM and other channel components.

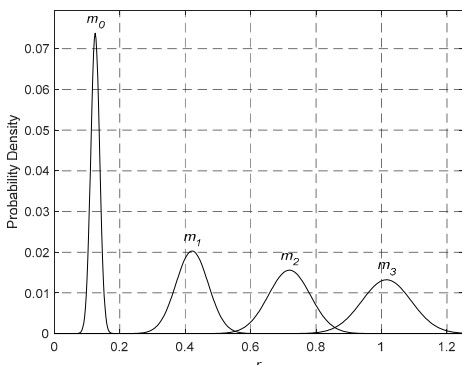

**Figure 1.** Conditional probability density functions (CPDF) of a PAM-4 signal, within signal-dependent noise.

### 3. Analog Optimization Method

It is well known that the input to output transfer function of optical modulators is usually non-linear. This non-linearity of the optical modulator can be used in a beneficial way to optimize the power spacing discussed above. Figure 2a presents a typical MZM L/V curve (light vs. applied voltage curve) which was achieved using the modulator DC bias sweep. The MZM DC bias point of the linear regime (quadrature point) in this example is 5 V, as shown in Figure 2a. Using a lower voltage bias point, of 4.1 V as an example, leads to operation within the non-linear regime. The eye diagrams of two different transmission schemes using two different bias points of 5 V and 4.1 V, in this example, are shown in Figure 2b,c, respectively. In Figure 2c, reducing the bias point to 4.1 V increases the spacing between level 3 and level 2 on account of reducing the spacing between level 1 and level 0, and increases the extinction ratio (ER). As will be shown in the following experimental section, the level spacing optimization leads to performance improvement by means of optical signal-to-noise ratio (OSNR) sensitivity and BER, in systems which are dominated by signal dependent noise. It should be pointed out that operating at the lower bias points leads to operation in the MZM non-linear regime which, in turn, introduces distortions by the MZM L/V curve, as shown in Figure 2c. These distortions are mostly compensated by the MLSE at the post-detection stage, as demonstrated in the following section. The other mechanism of improvement is enabled in a transmission system which uses a booster optical amplifier with a fixed optical output power, which is the common practice in a long reach fiber optic application. The reduction of the MZM bias reduces the average optical power at the optical amplifier input which increases the optical amplifier gain and in turn, increases the outer OMA (OMAouter) at the optical amplifier output. Consequently, the power level spacing between all levels is uniformly increased.

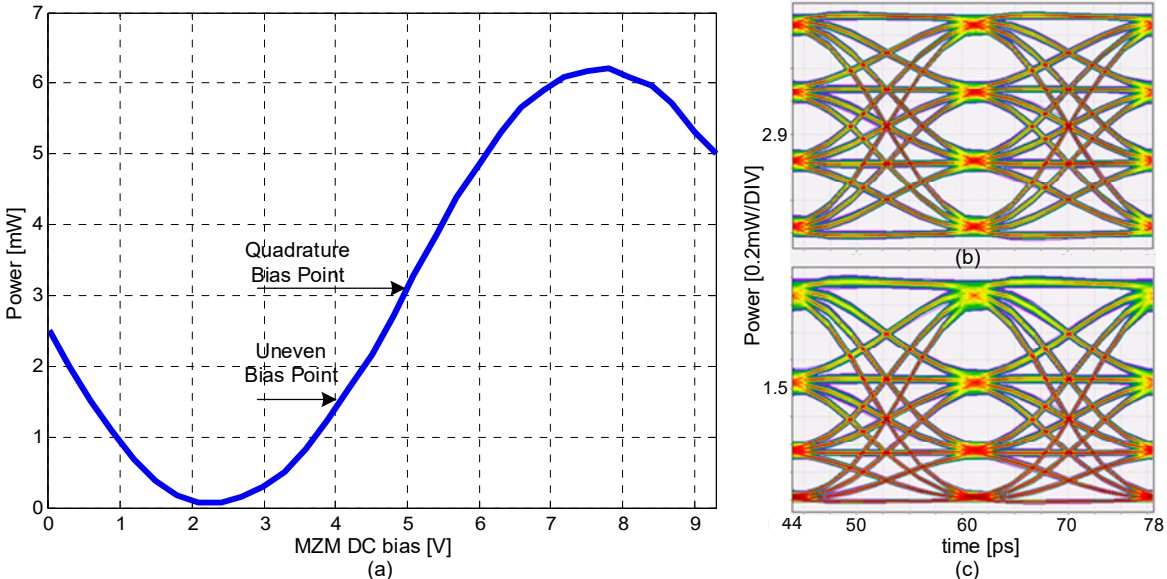

**Figure 2.** Eye diagrams at the optical transmitter output: (**a**) L/V curve of MZM; (**b**) even PAM at quadrature bias point of 5 V; (**c**) uneven PAM at bias point of 4.1 V.

### 4. Lab Measurements

#### 4.1. Lab Measurement Setup

A full optical link experiment was performed over standard single mode fiber (SSMF) and is presented in Figure 3.

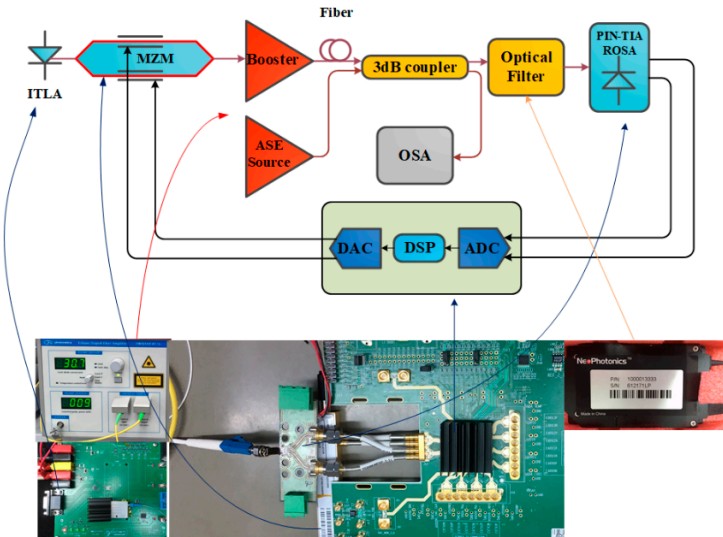

**Figure 3.** Lab measurement setup.

A 53 Gbaud PAM–4 signal is transmitted by a DAC with 3 dB cutoff frequency at 30 GHz and electrical output swing of 1 Vpp differential and the DAC output level is equally spaced. The DAC output is converted to the optical domain by an external MZM (Fujitsu FTM7937EZ) with Vpi DC and Vpi RF of 5.5 V and 3 V, respectively. The MZM is driven by integrated transmitter laser assembly (ITLA). The MZM optical signal output is amplified by a booster erbium-doped fiber amplifier (EDFA) configured to a fixed output power of 6 dBm, passes through optical fiber of 2 km length, representing a residual dispersion, and is accompanied by amplified spontaneous emission (ASE) noise using a 3-dB optical coupler and ASE source, in order to generate controlled OSNR conditions. Following, the optical signal passes through a 75 GHz (full width) 3 dB optical band pass filter and is converted to the electrical domain by a receiver optical sub assembly (ROSA-Semtech GN3289). The electrical signal is sampled and quantized by a 53 GSample/sec ADC with a 3-dB cutoff frequency at 19 GHz and ENOB of 6. The OSNR is measured at the optical coupler output using optical spectrum analyzer, and it is varied by controlling the ASE source while the average optical input power at the receiver is kept at $-1$ dBm. The bandwidth of the overall optoelectronic channel is 12 GHz (3 dB point).

Consequently, substantial ISI is introduced and advanced real time processing is applied. The received samples are equalized and processed by a MLSE DSP [3], which includes a feed forward equalizer with 15 taps followed by an MLSE equalizer with a memory depth of 3. The resulting measured BER curves are presented in Figure 4a,b.

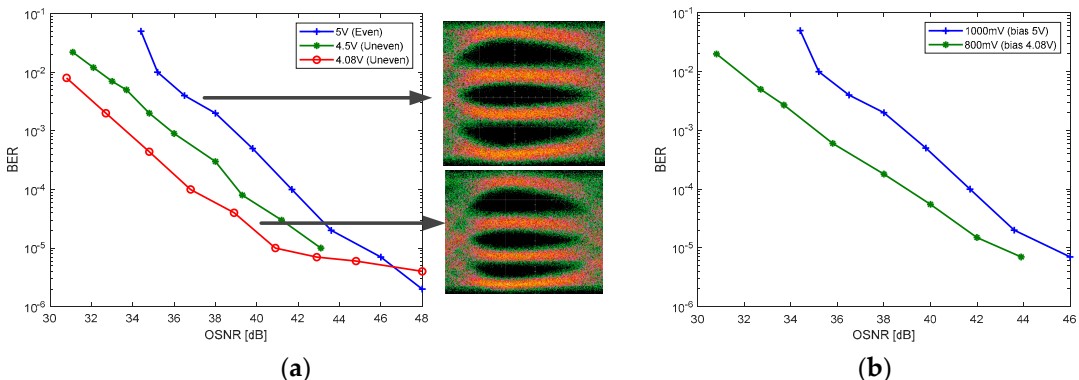

(**a**)　　　　　　　　　　　　　　　　　　　　　　　　(**b**)

**Figure 4.** BER vs. OSNR: (**a**) BER vs. OSNR for different MZM bias values. (**b**) Even power levels spacing at bias of 5 V, (blue curve '+') compared to uneven power levels spacing with bias of 4.08 V, (green curve '*'), while keeping identical OMAouter using 1000 mV and 800 mV MZM input voltage swing.

*4.2. Experimental Results*

In Figure 4a, three BER curves vs. OSNR are presented, each one associated with different MZM bias points. The optimization was performed by choosing the optimal modulator bias which obtains the lowest BER at OSNR of 38 dB. The '+' (blue), '*' (green) and 'o' (red) curves show the performance of quadrature point of 5 V bias (PAM-4 levels are equally spaced), and uneven PAM bias points of 4.5 V and 4.08 V, respectively. The performance improvement of the '*' (green) and the 'o' (red) curves is associated with the uneven PAM transmission and the increase of OMAouter, by using MZM bias points below quadrature. The OMAouter is defined as the difference between the power of level '3' and the power of level '0'. Since the booster EDFA output (Paverage) is configured to a fixed output optical average power of +6 dBm, the reduction of MZM bias reduces the average power at the EDFA input which increases the EDFA gain and in turn, increases the OMAouter at the EDFA output. The OMAouters of the curves marked with '+' (blue), '*' (green) and 'o' (red) shown in Figure 4a are 650 μW, 780 μW and 840 μW, respectively. Consequently, the OSNR improvement of the '*' (green) curve compared to the '+' (blue) curve (quadrature bias point) due to higher OMAouter is roughly 0.8 dB (10·log (780/650)). On the other hand, Figure 4a reveals that the overall OSNR sensitivity improvement of the '*' (green) curve compared to the '+' (blue) curve is 3 dB (at BER of 1e-3). The additional OSNR sensitivity improvement of 2.2 dB can be explained by the uneven PAM modulation in the case of the '*' (green) curve conditions, contributing better immunity to the multiplicative signal-spontaneous (signal dependent) noise at the higher PAM levels, leading to additional OSNR performance improvements. Similarly, the signal improvement of the 'o' (red) curve vs. the '+' (blue) curve due to higher OMAouter is roughly 1.1 dB (10·log (840/650)), while the overall OSNR sensitivity improvement of the red curves compared to the '+' (blue) curve is 5.3 dB. Again, this can be explained by the uneven PAM transmission which leads to additional 4.2 dB of OSNR sensitivity improvement.

Also observed in Figure 4a is the fact that in the case of bias of 4.08 V, the BER reduces slowly at large OSNR values (larger than 41 dB). This is due to the fact that under large OSNR conditions, while operating the MZM at low bias point, the dominant BER limiting factor becomes the nonlinearly of the transmitted signal. On the other hand, under lower OSNR conditions, the multiplicative ASE noise dominates, and the proposed uneven PAM scheme outperforms the "classical" MZM operation at its quadrature point.

Figure 4b presents two BER curves vs. OSNR, where each one is related to a different MZM bias point and a different MZM input voltage swing, such that both curves introduce the same OMAouter of 650 μW. It is shown that the bias point of 4.08 V and 800 mVpp voltage swing (green curve '*') outperforms the quadrature bias point of 5 V and 1 Vpp swing (blue curve '+') by 4 dB.

In this experiment, the improvement is exclusively related to the uneven power levels spacing introduced by the lower bias point, as both curves were generated under identical OMAouter conditions. This value of 4 dB is in a very good agreement with the predicted improvement of the same bias point introduced by the uneven PAM-4 case discussed in Figure 4a. It should also be noted that the signal-dependent ASE noise reduces as the OSNR increases, essentially decreasing the improvement of the uneven PAM transmission.

The uneven PAM-4 net gain (OSNR improvement) at BER of 1e-3 versus various bias points is presented in Figure 5a. A clear monotonic performance improvement versus uneven bias point is observed, manifesting the advantage of the proposed optical pre-distortion scheme.

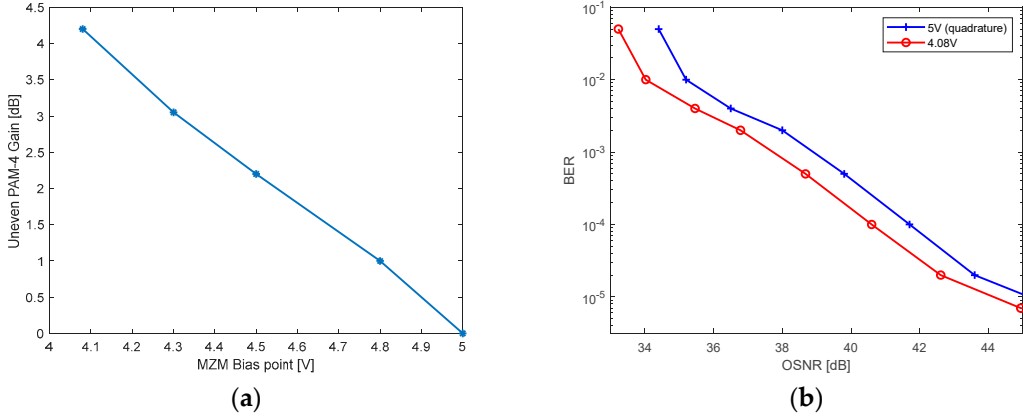

**Figure 5.** (**a**) Uneven PAM-4 gain vs. MZM bias point. (**b**) OMAouter improvement (only), by keeping power levels spacing of both curves identical.

Figure 5b presents two BER curves vs. OSNR, where the '+' (blue), and 'o' (red) are related to 5 V bias point (quadrature) and 4.08 V bias point, respectively. In this experiment, the two inner power levels (1 and 2) of the 4.08 V bias point (red curve) was configured to even power levels spacing such that both curves are under identical power levels spacing and only the OMAouter improvement is introduced. It is shown that the 4.08 V bias point outperform the quadrature bias point by ~1 dB. This value of ~1 dB is in a good agreement with the estimated improvement of the OMAouter introduced by the 4.08 V (uneven) PAM-4 case discussed in Figure 4a.

## 5. Conclusions

An analog–optical pre-distortion of MZM operation is proposed. Two mechanisms of benefit are achieved by manipulating the MZM bias point. Optimal spacing between the four levels of the PAM-4 signal can be achieved according to the statistics and distribution of the various noise ingredients, particularly in multiplicative noise environments where ASE noise dominates. In addition, allowing operation of the MZM in its non-linear transmission regime enables increased OMA. In turn, the non-linear distortions are post-compensated by a non-linear equalizer, such as an MLSE. Experimental measurements reveal that utilizing the proposed optical pre-distortion scheme in PAM-4 transmission leads to OSNR sensitivity improvement of 5.3 dB at pre-FEC BER value of 1e–3.

## 6. Patents

Analog Optical Pre-Distortion of PAM-4 and High Order Modulated Signals, Application number 15/242,502.

**Author Contributions:** Conceptualization, E.S. and G.K.; Methodology, E.S. and G.K.; Software, E.S. and G.K.; Validation, E.S. and G.K. and D.S.; Formal Analysis, E.S. and G.K.; Investigation, E.S. and G.K.; Resources, E.S., G.K. and D.S.; Data Curation, E.S. and G.K.; Writing-Original Draft Preparation, E.S. and G.K.; Writing-Review & Editing, E.S., G.K. and D.S.; Visualization, E.S., G.K. and D.S.; Supervision, D.S.

**Funding:** This research received no external funding.

**Conflicts of Interest:** The authors declare no conflict of interest.

## Appendix A

*Appendix A.1 Analytical Derivation of PAM-4 Signal with Optimum Transmission Levels*

In PAM-4 transmission, due to noise, the received signal *r* may be incorrectly classified, which in turn results in decision errors given by:

$$P_e = \frac{1}{4}P(e/0) + \frac{1}{4}P(e/1) + \frac{1}{4}P(e/2) + \frac{1}{4}P(e/3) \tag{A1}$$

where $P(e/l)$, $0 \leq l \leq 3$ is the conditional probability of error given that symbol $l$ was transmitted.

*Receiver optimization:* Assuming that the conditional probability density functions (CPDFs) are Gaussian and that the decision threshold is optimized to achieve minimum BER, the probability of error can be expressed by:

$$P_e = \frac{1}{8}erfc\left(\frac{I_1-I_0}{\sigma_1+\sigma_0}\right) + \frac{1}{8}\left[erfc\left(\frac{I_1-I_0}{\sigma_1+\sigma_0}\right) + erfc\left(\frac{I_2-I_1}{\sigma_2+\sigma_1}\right)\right]$$
$$+ \frac{1}{8}\left[erfc\left(\frac{I_2-I_1}{\sigma_2+\sigma_1}\right) + erfc\left(\frac{I_3-I_2}{\sigma_3+\sigma_2}\right)\right] + \frac{1}{8}erfc\left(\frac{I_3-I_2}{\sigma_3+\sigma_2}\right) \tag{A2}$$

where *erfc* is the complementary error function, $I_l$ ($0 \leq l \leq 3$) is the received signal level assuming symbol $s_l$ (level $l$) was transmitted, $\sigma_l$ ($0 \leq l \leq 3$) is the noise variance associated with each received level $I_l$.

*Transmitter optimization:* minimum BER is obtained by manipulating the transmitted levels of the PAM-4 signal. The optimization scheme is achieved by maintaining all the six *erfc* terms of Equation (A2) identical. The resulting optimization scheme is expressed mathematically by [6]:

$$\frac{I_1 - I_0}{\sigma_1 + \sigma_0} = \frac{I_2 - I_1}{\sigma_2 + \sigma_1} \tag{A3}$$

and

$$\frac{I_2 - I_1}{\sigma_2 + \sigma_1} = \frac{I_3 - I_2}{\sigma_3 + \sigma_2} \tag{A4}$$

where $I_0$ and $I_3$ are the outer PAM-4 levels and are obtained by the average optical input power at the receiver input, photodiode responsivity and the signal voltage swing at the modulator input, and $I_1$ and $I_2$ are independent variables that form the PAM-4 level spacing optimization.

*Appendix A.2 Analytical Comparison of PAM-4 Levels Optimization with Equal Field Spacing Assuming Signal-Dependent Noise*

Assuming that the amplitudes of the optical field are equally spaced, we can introduce the received signal levels such that:

$$I_0 = A^2,\ I_1 = (A + d)^2,\ I_2 = (A + 2d)^2,\ I_3 = (A + 3d)^2 \tag{A5}$$

where $A$ ($A = \sqrt{I0}$) is the amplitude of optical field associated with level 0 and $d$ is the optical field spacing. Substituting Equation (A5) into Equation (A3) yields:

$$\frac{(A + d)^2 - A^2}{\sigma(A + d) + \sigma A} = \frac{(A + 2d)^2 - (A + d)^2}{\sigma(A + 2d) + \sigma(A + d)} \tag{A6}$$

where $\sigma$ is the variance of the optical noise.

The result of both sides of Equation (A6) is $\frac{d}{\sigma}$, implying that the equality of Equation (A3) is met, i.e., the optimization requirement of Equation (A3) is satisfied.

Similarly, substituting Equation (A4) into Equation (A5) yields the same result $\frac{d}{\sigma}$, i.e., implying that the equality of Equation (A4) is also met.

Thus, equal optical field spacing introduces minimum BER assuming that signal-dependent noise dominates.

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
