# Peer review of "MZM Optimization of PAM-4 Transmission in Data Center Interconnect"

_applsci, doi:10.3390/app9040637_

Round 1
Reviewer 1 Report
The manuscript APPLSCI – 420992 proposes and demonstrates two mechanisms of improving PAM-4 transmission: the first one, by means of an optimal spacing between the four levels of the PAM-4 thanks to lower operation bias in the MZM; and the second one, increasing the difference between the power level ‘3’ signal and the power level of ‘0’ signal at the booster OSA output, allowing operation of the MZM in its non-linear transmission regime. The ideas are original and very interesting.
Results of OSNR sensitivity improvement of 5.3dB lead to a direct application in Data Center Interconnections. The presented ideas have been protected by the application of the corresponding patent.
Some typographical errors have to be corrected in pag.1: “optimization” in line 11; and, “recent” in line 24.
For these reasons, I recommend:
ACCEPT AFTER MINOR REVISION (CORRECTIONS TO MINOR METHODOLOGICAL ERRORS AND TEXT EDITING)
Author Response
Dear reviewer,
We appreciate the reviewers' review and efforts of our paper.
Response to Reviewer 1 Comments:
Point 1: Some typographical errors have to be corrected in pag.1: "optimization" in line 11; and, "recent" in line 24.
Response 1: The typographical errors are corrected in both lines 11 and 24 to: "optimization" and "recent", accordingly.
Sincerely,
Mr. Eduard Sonkin
Reviewer 2 Report
Dear editors and authors,
in the manuscript "MZM Optimization of PAM-4 Transmission in Data Center Interconnect" the authors propose and analyze analog optimization of 4-level pulse amplitude modulation (PAM-4) signals; the optimization is demonstrated experimentally showing a 4-5dB improvement in the signal-to-noise sensitivity. Specifically, the authors present and discuss two optimization schemes.
The work is quite interesting and I recommend the manuscript for publication. Before doing so, I only would like to raise two minor cosmetic comments:
(1) The left hand side of equation (1) depends on the index l=0,...,3, whereas the right hand side is independent of l. Is the right hand side really independent of l, as the equation suggests? Can the authors add appropriate indices to the terms that depend on the index l?
(2) There are still some typos. For example, in line 24 "resent" -> "recent".
Author Response
Dear review,
We appreciate reviewers' review and efforts on our paper.
Response to Reviewer 2 Comments:
Point 1: The left hand side of equation (1) depends on the index l=0,...,3, whereas the right hand side is independent of l. Is the right hand side really independent of l, as the equation suggests? Can the authors add appropriate indices to the terms that depend on the index l?
Point 2: There are still some typos. For example, in line 24 "resent" -> "recent".
Response 1: The equation (1) is corrected by adding the appropriate indices l at the right hand. Please find the attached paper with corrections by "Track changes".
Response 2: The typographical error is corrected in line 24 to: "recent".
Sincerely,
Mr. Eduard Sonkin
